# Thermosensitive Polyester Hydrogel for Application of Immunosuppressive Drug Delivery System in Skin Allograft

**DOI:** 10.3390/gels7040229

**Published:** 2021-11-23

**Authors:** I-En Wu, Madonna Rica Anggelia, Sih-Yu Lin, Chiao-Yun Chen, I-Ming Chu, Cheng-Hung Lin

**Affiliations:** 1Department of Chemical Engineering, National Tsing Hua University, Hsinchu 300, Taiwan; k2258s91@icloud.com (I.-E.W.); jolin552283@gmail.com (S.-Y.L.); chiaoyun.chen@rochester.edu (C.-Y.C.); 2Center for Vascularized Composite Allotransplantation, Department of Plastic and Reconstructive Surgery, Chang Gung Memorial Hospital, Chang Gung Memorial Medical College, Chang Gung University, Taoyuan 333, Taiwan; mranggelia@yahoo.com or

**Keywords:** thermosensitive, tacrolimus, allotransplantation

## Abstract

Tacrolimus (FK506) is a common immunosuppressive drug that is capable of suppressing acute rejection reactions, and is used to treat patients after allotransplantation. A stable and suitable serum concentration of tacrolimus is desirable for better therapeutic effects. However, daily drug administration via oral or injection routes is quite inconvenient and may encounter drug overdose or low patient compliance problems. In this research, our objective was to develop an extended delivery system using a thermosensitive hydrogel of poly ethylene glycol, D,L-lactide (L), and ϵ-caprolactone (CL) block copolymer, mPEG-PLCL, as a drug depot. The formulation of mPEG-PLCL and 0.5% PVP-dissolved tacrolimus was studied and the optimal formulation was obtained. The in vivo data showed that in situ gelling is achieved, a stable and sustained release of the drug within 30 days can be maintained, and the hydrogel was majorly degraded in that period. Moreover, improved allograft survival was achieved. Together, these data imply the potential of the current formulation for immunosuppressive treatments.

## 1. Introduction

Hydrogels are formed by crosslinking hydrophilic polymers, either through chemical or physical interactions. They can be applied to many biomedical needs, such as drug delivery and tissue engineering [1,2,3,4,5,6,7,8]. Thermosensitive hydrogel, which can form gel in situ at the body temperature, is especially useful as a depot for the delivery of therapeutics with minimal invasiveness [9,10]. Therapeutic drugs or cells can be easily mixed in the sol-phase of the hydrogels before injection and be stably released from the gel-phase inside the body.

Thermosensitive hydrogels synthesized from poly (ethylene glycol)-polyester copolymers are suitable for biomedical applications due to their good biocompatibility and gel-forming properties [11]. The mild in-situ gelling reaction for ease of encapsulation and sustained release of therapeutic agents or cells [12] while the degradation rate of hydrogels can be modified to suit the particular applications by choosing proper ester monomer constituents. In our previous study [13], we successfully developed a thermosensitive mPEG-amino hydrogel for sustained release of a hydrophobic drug, which is tacrolimus. However, the degradation rates of this type of hydrogels were very slow, so the depot persisted more than 3 months. It raises concerns about fibrosis formation at the injected site [14,15]. Therefore, poly (ethylene glycol)-polyester hydrogel is investigated in this study. Thermosensitive hydrogel methoxy-poly(ethylene glycol)-co-poly(lactic acid)-poly(e-caprolactone), mPEG-PLCL, was examined for its suitability as a drug carrier.

Tacrolimus (FK506), produced from *Streptomyces tsukubaensis*, is one of the most common immunosuppressive drugs used to treat acute rejection after allotransplantation [16,17,18]. In general, tacrolimus is able to globally inhibit the gene expression in T cell by combining with FK binding protein 12 and forming a complex, namely tacrolimus-FKBP. The complex will inhibit the activity of nuclear factor of activated T-cells (NFAT) and further block the activation pathway and proliferation of T cells [19]. However, the strong hydrophobicity of tacrolimus presents challenges in formulating dosage forms and requires the use of a cosolvent to dissolve the drug prior to suspension in the sol-phase of the hydrogels [20,21].

PVP (Polyvinylpyrrolidone) is a polymer polymerized of N-Vinylpyrrolidone monomers which can be used as possesses the cosolvent because of its characteristics including several inertias, non-toxicity, temperature resistance, pH stability, and biodegradability. It can act as the cosolvent of hydrophobic and hydrophilic drugs [22]. Due to these advantages, PVP has wide applications in the fields of biomedical and food industries [23,24]. Therefore, PVP was chosen in our formulation study.

The developed hydrogel was then used to carry tacrolimus and the pharmacokinetic and efficacy were tested to demonstrate the feasibility of the hydrogel in skin allotransplantation in the rat model.

## 2. Results and Discussion

### 2.1. The Synthesis and Characterization of mPEG-PLCL

The synthesized mPEG-PLCL was characterized by 1H-NMR, GPC, and FTIR. The final product was transparent and light yellow in color, as shown in Figure 1A. The results of the ^1^H-NMR analysis are shown in Figure 1B. The molecular weights of each segment were calculated by GPC or NMR as tabulated in Table 1. By these calculations, the ratio of LA/CL was found to be at 78/22. The FTIR results showed the existence of a C=O bond at 1750 cm^−1^ and C-O bond at 1100 cm^−1^, respectively, proving that the synthesis was successfully implemented in Figure 1C. The molecular weights obtained were found to be consistent with the original fed ratios. The PDI (Polymer dispersity index, Mw/Mn) of the polymer was found to be 1.03, indicating a rather uniform distribution of the molecular weights.

### 2.2. Sol–Gel–Sol Transition

The sol-to-gel transition is an important phenomenon of thermosensitive hydrogels. Different weight percentages of the hydrogel will exhibit different phase-transitional behaviors, as the temperature changes. The hydrogel concentration ranged from 5 to 25 wt% by weight and the results are shown in Figure 2A. At concentrations of 5 and 10 wt% the solution failed to form a gel phase as the temperature increased. The sol-to-gel transition was only observed for samples of higher than 15 wt% concentration. A representative of phase transition was shown with the 25 wt% sample. The morphology of 25 wt% mPEG-PLCL at different temperatures is shown in Figure 2B. Under 10 °C, the polymeric mixture appears as a liquid solution, possibly in a micellar solution. As the temperature increases, hydrogen bonds between the polymeric units form a viscous transparent gel-like phase. However, as the temperature increased to over 22 °C, this gel-like phase became less viscous and exhibited fluid-like solution-phase behavior. The hydrogel was formed at 32 °C. Precipitation occurred at 46 °C. The viscosity of 25 wt% mPEG-PLCL at different temperatures are shown in Figure 2C. The highest viscosity value was found between 10–20 °C, which reached around 200 dPa*s, indicating the existence of a very viscous gel-like phase within the temperature range, probably due to a certain transition between different micellar forms [25]. As the temperature increases, the interaction of hydrogen bonds weakens, leading to a less viscous, more fluid-like solution phase. The hydrogel phase was formed at 32 °C and the corresponding viscosity was about 20 dPa × s. Based on the above observations, the 25 wt% polymer concentration level was selected to be the formulation for further study, since it possesses suitable phase transition properties.

### 2.3. Cosolvent Selection and Encapsulation Efficiency

The encapsulation efficiency (EE) of tacrolimus was evaluated using different cosolvents. Cosolvent is necessary for the dissolution of tacrolimus before it is mixed with hydrogel solution. It is essential that tacrolimus is distributed uniformly in hydrogels as to have steadier release rates. All tests were performed with 25 wt% hydrogel samples. Table 2 indicates that all of the cosolvent tested achieved high encapsulation efficiency. It is noted that the amount of PVP used is rather low compared to those of the other two solvents. This indicates superior dissolution ability of PVP and low toxicity that may be caused by co-solvent in the formulation.

### 2.4. Biocompatibility Testing

To prove the biocompatibility of formulations involving various cosolvents, the MTT assay was conducted. We divided the samples into seven groups. The results are shown in Figure 3. Each group showed low cell toxicity, with approximately 80% on day 1 (Figure 3A). However, in the DMSO cosolvent group (Group B), the cell viability dropped to only 60% on day 2 (Figure 3B). DMSO seemed to accelerate the hydrolysis of the hydrogel, probably because of the increased acidity of the mPEG-PLCL degradation. PVP or ethanol cosolvent groups (groups A and C) showed the best biocompatibility. Therefore, PVP was chosen to be used in subsequent experiments.

### 2.5. In Vitro Degradation and Tacrolimus Release

The formulation of 25 wt% mPEG-PLCL with 0.5% PVP was tested with loaded tacrolimus at 10 mg/mL. The hydrogel was completely degraded in vitro during 30-day period, as shown in Figure 4A, where the polymer was converted to monomer size (Mn was less than 200) according to the results. Furthermore, sustained in vitro release of tacrolimus was demonstrated in vitro, as shown in Figure 4B. The lack of burst release indicates the ability of this formulation to uniformly dispersed tacrolimus inside the hydrogel by this formulation. In Figure 4C, surprisingly, the final cumulative releases of tacrolimus reached only 60% of the original loading, while most of the hydrogels were virtually degraded. This may be attributed to the different approaches used in the degradation and release tests. The drug release tests were conducted using 1000 Da dialysis cassettes, whereas the degradation tests were using micro-centrifuge tubes. Thus, small disintegrating hydrogel debris may be absent in the degradation tests, and may still be able to trap some tacrolimus inside the dialysis cassettes. However, the absence of burst release and the prolonged steady release of tacrolimus demonstrated the advantages of this formulation as a drug delivery vehicle.

### 2.6. Allograft Skin Transplantation

In the animal experiment, the hydrogel formulation significantly prolonged the allograft survival, as compared to the control (median survival time 19.5 days vs. 9 days, *p* = 0.001), Figure 5A. The blood concentration of tacrolimus, shown in Table 3, was stable for 30 days. The stable level of the drug maintained for an extended period of time is beneficial for the treatment and convenient to administer for better therapeutic effect.

An in vivo degradation of the hydrogels within 30 days can be seen in Figure 5B, where the smaller of hydrogel mass presence. Besides, two out of six recipients received mPEG-PLCL injection that showed accepted skin allograft at POD 30 demonstrated intact skin tissue, and healthy hair growth. No erythema, edema, or desquamation phenomena that accompany immune rejection were observed. In the rejected group and the untreated groups, the skin allograft showed epidermolysis, hair loss, and necrosis of the epidermis with severe, dense inflammation in the upper and deep dermis. (Figure 5B). Additionally, although there is no significant alteration in peripheral blood levels of CD4 T-cells, CD8 T-cells, and regulatory T cells, shown in Figure 5C, the percentages of CD4 T cells and CD8 T cells were lower and percentage of Tregs was higher compared to the untreated group at POD 14. It has been reported that tacrolimus affects T cells proliferation [26]. The insignificant results may indicate that the injection of a hydrogel loaded with tacrolimus showed more a local suppressive effect than a systemic effect. The critical goal for local drug injection in transplantation is to minimize the systemic side effect [27]. However, the blood concentration of tacrolimus with the current formulation is over 20 ng/mL, a level higher than the effective therapeutic level of 8–10 ng/mL for patient [28]. Thus, optimal dosage of tacrolimus with the current formulation should be further investigated to improve the drug efficacy.

## 3. Conclusions

The hydrogel-based tacrolimus delivery system developed in this study demonstrates various advantageous features, including (i) in situ gelling, (ii) no burst release, (iii) slow and steady release over extended period of time, (iv) improved allograft outcome, and (v) majorly degraded in 30 days. In this system, PVP is used in the formulation as a cosolvent for dissolution of tacrolimus prior to the mixing with hydrogel polymers. The amount PVP needed for uniform dispersion of tacrolimus is quite small and poses minimal harmful effects toward cells. This delivery system is easy to use and could potentially represent a novel approach to immunosuppression treatment after allotransplantation.

## 4. Materials and Methods

### 4.1. Synthesis of Methoxy Poly(ethylene glycol), D,L-lactide, and ϵ-caprolactone Block Copolymer (mPEG-PLCL)

The copolymer was synthesized by one-step ring-opening polymerization shown in Figure 6, with mPEG (MW = 550 Da, 6.35 g, 11.5 mmol) (Sigma-Aldrich, Inc., St. Louis, MO, USA), D,L-lactide (13.21 g, 91.6 mmol) (Purac Biochem, The Netherlands) and ϵ-caprolactone (3 g, 26.3 mmol) (Alfa-Aesar Chemical, ThermoFisher Scientific, Waltham, MA, USA) using 0.1 wt% stannous octoate (Sigma-Aldrich, Inc., St. Louis, MO, USA) as the catalyst. Before synthesis, each chemical and glassware were placed under vacuum overnight for moisture removal. mPEG was further dried by placing it inside a round bottle flask, pre-heated in a 150 °C oil-bath, with repetitive vacuum and nitrogen purge.

All materials were mixed in a reactor and gently stirred under a nitrogen environment at 150 °C for 6 h. After completion of the reaction, the product mixture was cooled and dissolved in 30 mL of chloroform (Avantor, Center Valley, PA, USA). The solution was then precipitated over 500 mL pre-cooling hexane (Echo Chemicals, Toufen, Miaoli, Taiwan). The precipitate was allowed to stand for 20 min and dissolved again in approximately 30 mL of dimethyl sulfoxide (DMSO) (Sigma-Aldrich, Inc., St. Louis, MO, USA). The solution was dialyzed with 1000 Da MWCO dialysis bag (Spectra/Por^®^ 6 Dialysis membrane, P.N 132640) for 7 days and lyophilized. The final lyophilized product was stored in a vacuum atmosphere for future use.

### 4.2. 1H Nuclear Magnetic Resonance (1H-NMR) Spectroscopy

NMR spectrometer (Varian Unityinova 500 MHz NMR) at the Instrument Center of National Tsing Hua University was used to examine the composition of copolymers. The copolymers were dissolved in Chloroform-d (CDCl_3_) (Sigma-Aldrich, Inc., St. Louis, MO, USA) and assayed.

### 4.3. Gel Permeation Chromatography (GPC)

A GPC system (JASCO PU-4180/RI-4030) was used with a Shodex Ohpak SB-803 HQ column at a flow rate of 1 mL/min at 40 °C. Copolymers were dissolved in dimethyl-formamide (DMF) (Avantor, Center Valley, PA, USA) with 0.1 wt% of LiBr at the concentration of 1%. The molecular weights of the copolymers and PDI were determined relative to those of the PEG standards.

### 4.4. Fourier-Transformed Infrared Spectroscopy (FT-IR)

The functional group identification of mPEG-PLCL hydrogel was done by Fourier-transformed infrared spectroscopy (FT-IR). Hydrogel infrared spectra was performed using an FTIR spectrometer (NicoletTM iS50, Thermo Fisher Scientific, Waltham, MA, USA) equipped with an attenuated total reflectance (ATR) module. The lyophilized hydrogel was placed on the test plate. Spectra were collected at a frequency ranging from 650 to 4000 cm^−1^ 32 times, and the resolution was 1 cm^−1^. The existence of a C=O bond at 1750 cm^−1^ and C-O bond at 1100 cm^−1^ was used to collect information on secondary structures and conducted for chemical structure verification.

### 4.5. Sol–Gel Transition Phase

The sol-to-gel transition behavior of mPEG-PLCL prepared at 5 to 25 wt% concentrations was investigated using a vial inverted method in 2 mL test tube at a temperature range of 10–50 °C with a temperature increment of 1 °C via dry bath incubator. The sample was allowed to equilibrate for 10 min at each temperature. The gel state was determined when the solution stopped flowing while being inverted and gently agitated.

### 4.6. Rheological Property of mPEG-PLCL

A rheometer (AR2000ex system, TA instrument) installed with 25 mm parallel plate and gap of 1mm was employed to measure the viscosity of the synthesize copolymer with increasing temperature. First, 25 wt% of mPEG-PLCL hydrogel solution was prepared. Approximately 500 μL of sample was loaded to the gap between the rheometer and parallel plate without any air bubble. Before starting, the stress rate at 1 rad/s and the power at 0.5% was set up. The temperature increased from 4-50 °C in 15 min and 1500 plots were collected. The viscosity of mPEG-PLCL with increasing temperature was recorded.

### 4.7. Quantification of Tacrolimus

To quantify the concentration of tacrolimus HPLC (Hitachi, D2000) with the column (COSMOSIL Packed Column, 5C18-AR-ii, 4.6 mm × 250 mm) was conducted. Acetonitrile (Sigma-Aldrich, Inc., St. Louis, MO, USA) was used as the mobile phase at a flow rate of 0.5 mL/min with a detector set at 213 nm.

### 4.8. Development of Formulation for Higher Encapsulation Efficiency

The optimal formulation of mPEG-PLCL was designed for tacrolimus encapsulation. Since tacrolimus is insoluble in water, a solvent that can dissolve tacrolimus to enable its dispersion in the polymer-containing solution was investigated. Ethanol or DMSO is commonly used for this purpose. However, these solvents, are harmful to cells at high concentrations. In our previous study, ethanol used as a cosolvent induced a certain degree of cytotoxicity in our in vitro test. Therefore, alternative solvents were tested in this study.

PVP-K30 (Polyvinyl Pyrrolidone K30) was found to be a suitable solvent for this purpose. In the test, 20 mg/mL of tacrolimus (LC Laboratories, Woburn, MA, USA) was dissolved by 0.5% of PVP-K30 (Emperor Chemical Co., LTD, Taiwan) and mixed with 25 wt% of mPEG-PLCL. For comparison, 10% ethanol and DMSO were prepared with the same polymeric solution. After mixing for overnight, the different formulations of hydrogel with tacrolimus were place in 37 °C for gelling. The formed gels were then rinsed with PBS and RO water and lyophilized. The encapsulation efficiency was measured for each formulation by HPLC.

### 4.9. In Vitro Degradation Testing of mPEG-PLCL

The biodegradability of the drug carrier was verified in vitro. The procedure is described below. Hydrogels of 25 wt% of mPEG-PLCL with 0.5% PVP-K30 were prepared. After gelling at 37 °C, samples were incubated in 2 mL micro centrifuge tube with 200 µL of hydrogel each. Then 1 mL of PBS was added as the release buffer to the tube, which was placed in a rotary incubator at 60 rpm at 37 °C. The PBS buffer was refreshed each two day and pellets were collected at different time points (Day 0, 5, 7, 9, 15, 20, 23, 27, 31). These pellets were lyophilized and dissolved in a solvent composed of DMF/0.1% LiBr for GPC measurement to assay the molecular weight of the remaining mPEG-PLCL hydrogel.

### 4.10. In Vitro Drug Release Testing of mPEG-PLCL

Hydrogels which were made of 25 wt% of mPEG-PLCL with 0.5% PVP-K30 and 10 mg/mL tacrolimus were tested. After mixing, 500 μL of the samples was added to 2000 Da cut-off dialysis cassettes (Slide-A-LyzerTM G2 Dialysis Cassettes, ThermoFisher Scientific, Waltham, MA, USA). The dialysis cassettes were placed in 25 mL of release buffer composed of PBS and 2% of Tween-20, at 37 °C. Samples of release buffer at different days (Day 0, 1, 5, 7, 9, 12, 15, 19, 23, 27, 31) were collected. The concentration of tacrolimus in the samples was measured by HPLC.

### 4.11. Biocompatibility Testing

Biocompatibility of various formulation was tested using MTT assay. The experimental groups are shown in Figure 7. The hydrogels were put into the insert of the transwell device, where HEK cells grew on the bottom of the device. In addition, the growth of cells under different cosolvents was compared with that without hydrogels. The MTT assay was performed as described below. The MTT reagent was prepared using 5 mg/mL of MTT agent (3-(4,5-Dimethyl-2-thiazolyl-2.5-diphenyl-2H-terazolium bromide)) in PBS. Subsequently, 50 μL of MTT agent was added into each well containing cells and incubated for 4 h. After the period, DMSO was added to stop the reaction, and a purple color appeared. One hundred microliters of medium was then collected and placed in a 96-well plate and the OD 550 nm was evaluated using an ELISA reader (Bio-Tek Synergy HT). Cell viability was calculated by the ratio of OD measurement to control.

### 4.12. Allograft Skin Transplantation Experiment

Male Brown-Norway (BN) and Lewis (LEW) rats, 8–12 weeks and weight 300–350g, were obtained from the National Laboratory Animal Center and BioLasco, Taiwan respectively. Animals were housed under pathogen-free conditions at the Animal Center of Linkou Chang Gung Memorial Hospital, according to Institutional Animal Care and Use Committee of Chang Gung Memorial Hospital protocols (IACUC No. 2017121807, approved date, 14 May 2018). A 1 cm × 1 cm of tail-skin graft from BN rat was transplanted onto the dorsal thorax of LEW rat (*n* = 6). After skin grafting, 1 mL of mPEG-PLCL/0.5% PVP hydrogel that was loaded with 10 mg of tacrolimus (Tac) was injected subcutaneously at dorsal site near allograft Figure 5B. Antibiotics (50 mg/kg cefazolin (Taiwan Biotech Co. Ltd., Taoyuan, Taiwan) and analgesics (2 mg/kg ketoprofen (Nang Kuang Pharmaceutical Co. Ltd., Tainan, Taiwan) were also injected during 3 post-operative days (POD) for post-operative care. One ml blood from each Lewis Rat was collected at POD 1, 7, 14, and 28 for tacrolimus determination by HPLC and peripheral blood analysis by flow cytometry. The rats’ biopsies were done at POD 30, and the degradation of hydrogel was observed. Furthermore, histology of the tissue at the injection site and skin allograft were subjected to Hematoxylin-Eosin (HE) staining.

### 4.13. Quantification of Circulating CD4 T Cells, CD8 T Cells, and Regulatory T Cells (Tregs)

To confirm the immunosuppressive effect of tacrolimus, the quantification of CD4 T cells, CD8 T cells, and Tregs was performed as described previously [29] using FACS Canto II flow cytometer (BD Bioscience, San Jose, CA, USA). Briefly, blood was collected from rat tail vein for analysis. Red blood cells were first lysed with ammonium–chloride–potassium (ACK) (Sigma-Aldrich, Inc., St. Louis, MO, USA) buffer at room temperature. Twenty million cells/mL of peripheral blood cells were stained with specific surface markers antibodies. Unspecific binding was blocked using 0.1% BSA (Sigma-Aldrich). TCR, CD4, CD8, CD25, and FoxP3 fluorescence conjugated antibodies were used to detect CD4 T cells, CD8 T cells, and Tregs. Prior to intracellular maker (FoxP3) staining, permeabilization buffer (BD Bioscience) treatment was performed. Leukocyte populations, such as lymphocytes, monocytes, and granulocytes were distinguished via forward scatter (FSC) and side scatter (SSC). All antibodies were purchased from BioLegend (San Diego, CA, USA) and BD bioscience.

### 4.14. Histology Assessment

Skin allograft and tissue at injection site biopsy specimens were formalin-fixed (Merck KGaA, Darmstadt, Germany). Five μm thickness was briefly prepared from paraffin-embedded specimens and sliced. After deparaffinization, and rehydration, specimens were stained with hematoxylin and eosin (HE) (Leica, Buffalo Grove, IL, USA). Slides were mounted and observed under light microscopy with 100x magnification.

### 4.15. Statistical Significance Analysis

Significant differences for the continuous variable were evaluated with unpaired two-tailed Student’s t-test using GraphPad Prism 9 (GraphPad Software, San Diego, CA, USA). Allograft survival was analyzed using the Kaplan–Meier method. Log-rank test was performed for pairwise comparison. Statistical significance was set at *p* < 0.05.

## Figures and Tables

**Figure 1 gels-07-00229-f001:**
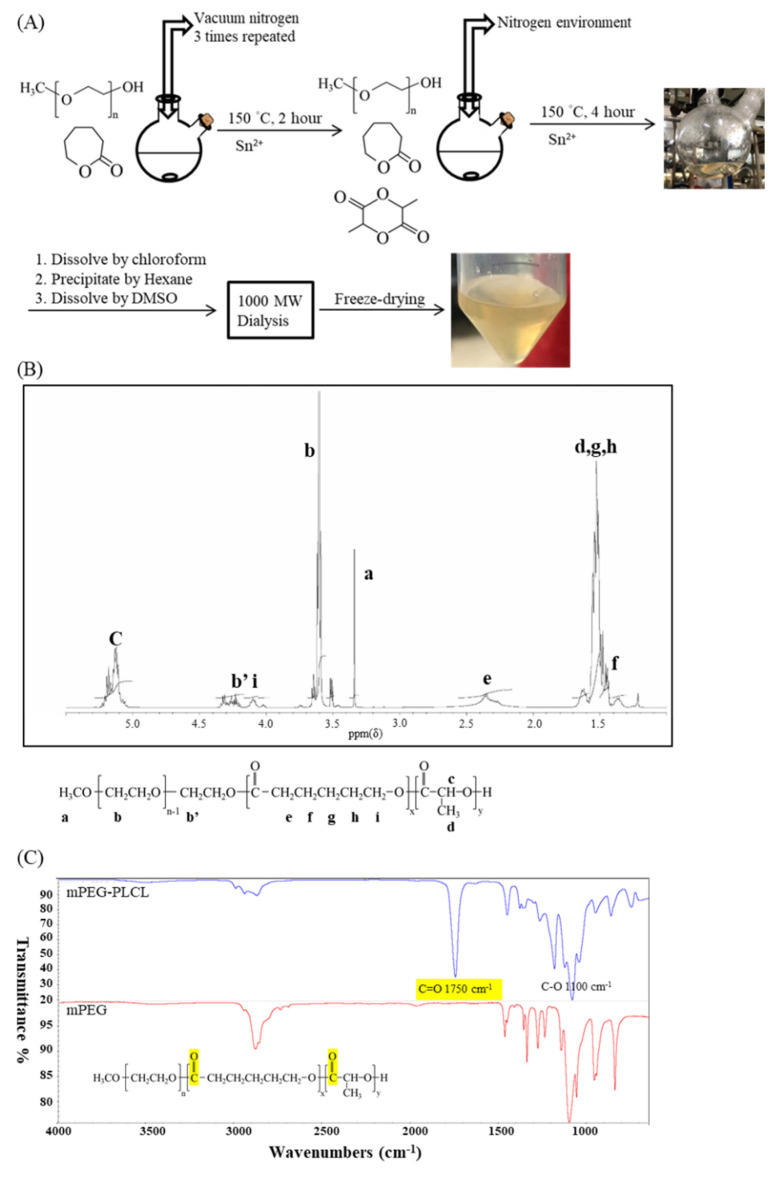
Synthesis and chemical properties of mPEG-PLCL. (**A**) Scheme of mPEG-PLCL synthesis. (**B**) 1H-NMR spectra. (**C**) FTIR spectra. The practical molecular weight was determined by the spectra of NMR in (**B**). n = 11, x = 2, y = 18. In (**C**), the highlighted part represents C=O bond signals.

**Figure 2 gels-07-00229-f002:**
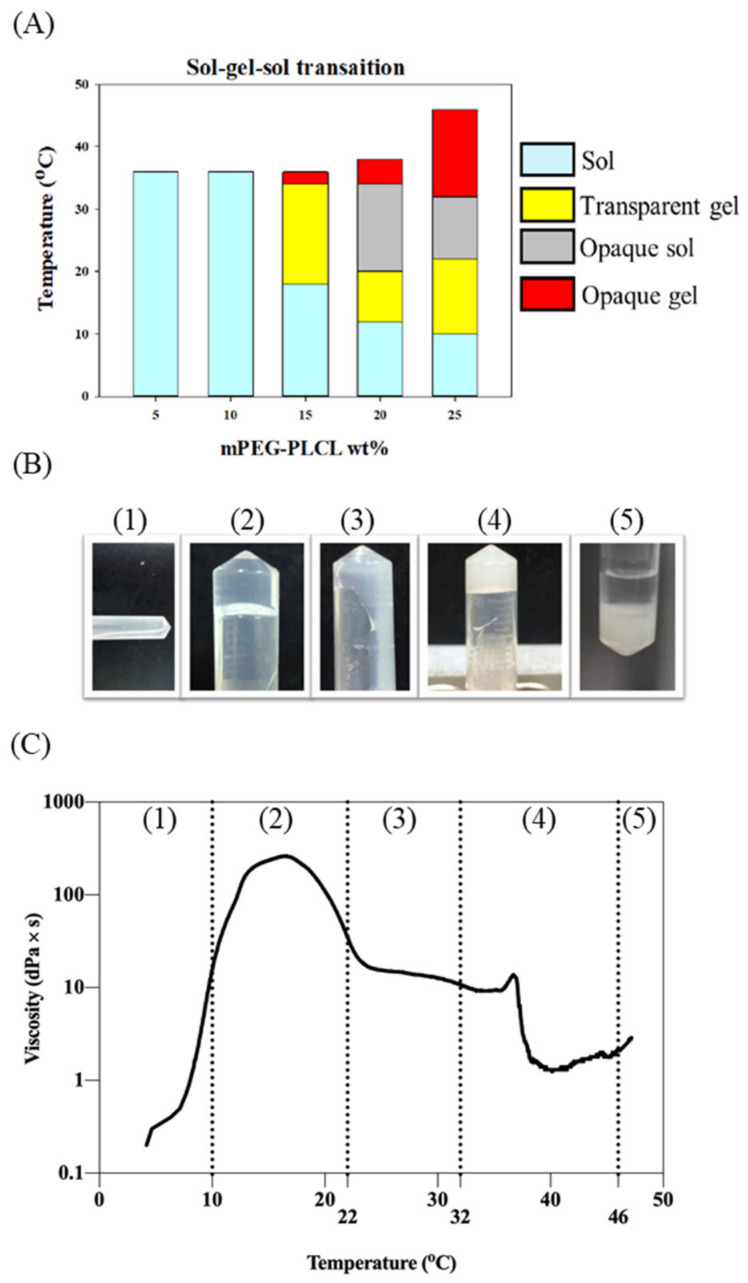
Phase transition with increasing temperature. (**A**) Sol-gel-sol transition testing of different concentration of mPEG-PLCL hydrogel. (**B**) Appearance of mPEG-PLCL at different temperatures. (**C**) Viscosity of 25 wt% mPEG-PLCL hydrogel with increasing temperature.

**Figure 3 gels-07-00229-f003:**
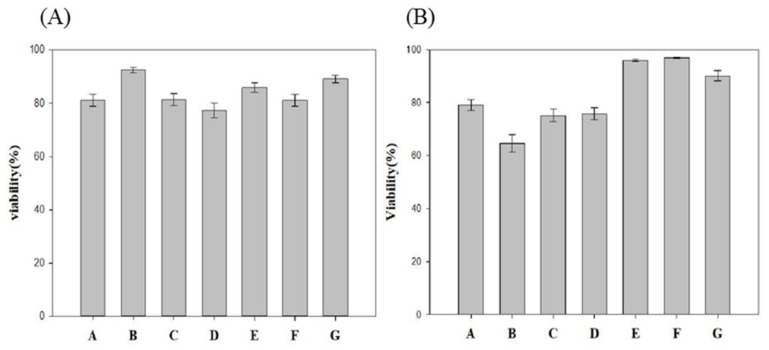
Biotoxicity testing using the MTT assay of different cosolvents with 25 wt% mPEG-PLCL on (**A**) Day 1. (**B**) Day 2.

**Figure 4 gels-07-00229-f004:**
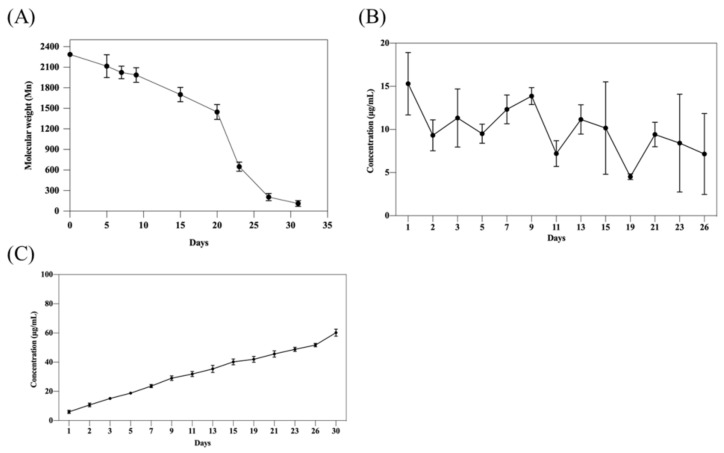
In vitro testing of mPEG-PLCL/0.5% PVP for (**A**) degradation. (**B**) Daily measurements of tacrolimus release. (**C**) Accumulated amount of tacrolimus release.

**Figure 5 gels-07-00229-f005:**
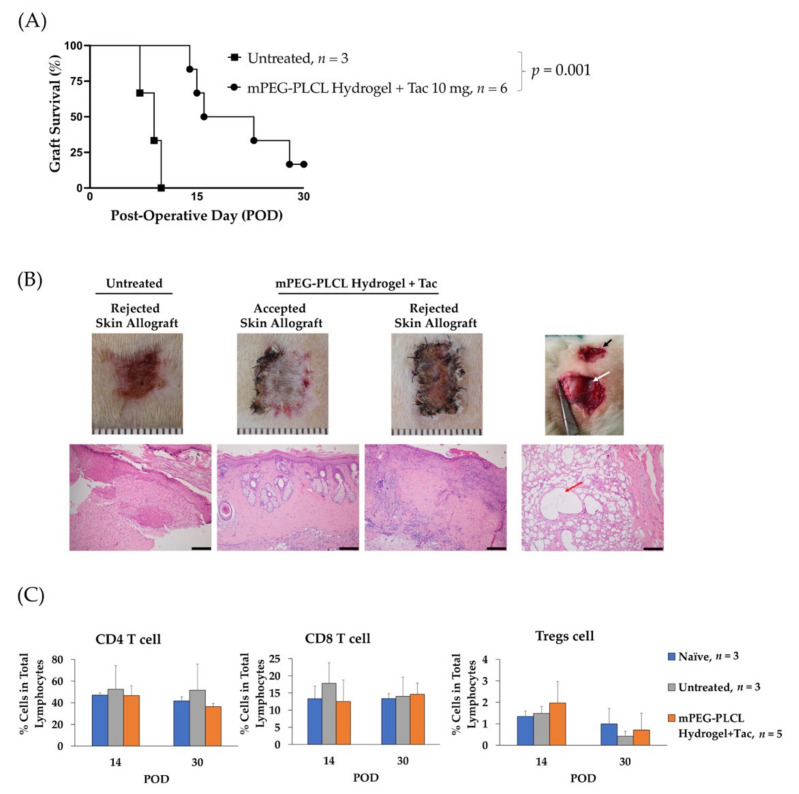
mPEG-PLCL hydrogel injection prolonged skin allograft on Lewis Rats. (**A**) Graft survival rate. (**B**) Macroscopic and microscopic view of skin allograft and tissue surrounded hydrogel. The black arrow indicates the location of the skin allograft. The white and red arrows indicate degraded hydrogel. Magnification: 100×, Scale bar: 200 µm (**C**) Peripheral blood (CD4 T cell, CD8 T cell and regulatory T cell) quantitative analysis. The Naïve and Untreated represent the level of peripheral blood from Lewis rats without transplantation and which received transplantation without hydrogel injection, respectively.

**Figure 6 gels-07-00229-f006:**
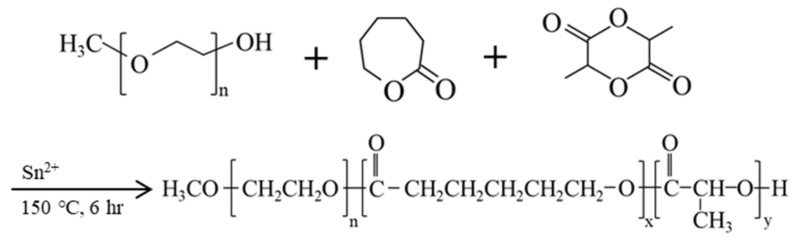
Synthetic scheme of mPEG-PLCL copolymer hydrogel by ring-opening polymerization of e-caprolactone and D,L-lactide, initiated by mPEG with Sn(Oct) 2 as a catalyst.

**Figure 7 gels-07-00229-f007:**
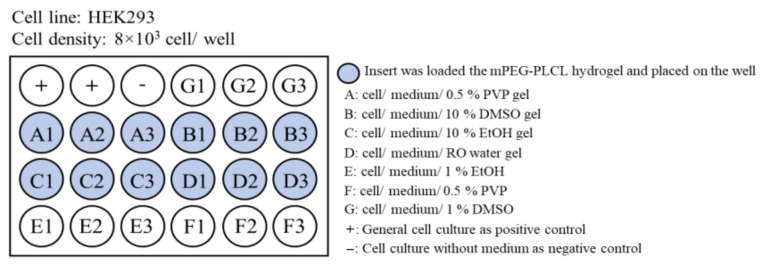
Biocompatibility testing of formulation of mPEG-PLCL hydrogel with different cosolvents and the effects of the cell growth.

**Table 1 gels-07-00229-t001:** Physical parameters of mPEG-PLCL.

Feed Molar Ratio of mPEG: LA: CL		1H-NMR	GPC		
		^a^ Mn	^b^ Mn	^b^ Mw	^c^ PDI
mPEG-PLCL	1:8:2	2042	1980	2034	1.03

^a^: Number-average molecular weight (Mn) determined by 1H-NMR; ^b^: Number-average molecular weight (Mn) and weight-average molecular weight (Mw) analyzed by GPC; ^c^: Polymer dispersity index (PDI) analyzed by GPC.

**Table 2 gels-07-00229-t002:** Encapsulation Efficiency (EE%) of Tacrolimus with different cosolvent in 25 wt% of mPEG-PLCL where 10 wt% of FK506 was loaded.

25 wt% mPEG-PLCL with	10% EtOH	10% DMSO	0.5% PVP
Encapsulation efficiency	98 ± 0.5%	95 ± 0.3%	99 ± 0.1%

**Table 3 gels-07-00229-t003:** Whole blood concentration of tacrolimus: In vivo drug release results from mixed hydrogels containing 10 mg/mL tacrolimus.

Time (Days)				
Tacrolimus (ng/mL)	1 24.7 *±* 3.8	7 42 *±* 7.4	14 29.4 *±* 5.9	21 26.05 *±* 3.8

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
