# Peer review of "Thermosensitive Polyester Hydrogel for Application of Immunosuppressive Drug Delivery System in Skin Allograft"

_gels, 2021, doi:10.3390/gels7040229_

Round 1

Reviewer 1 Report

                                                   Reviewer Recommendation 

Manuscript Title: “Thermosensitive polyester hydrogel for application of immunosuppressive drug delivery system in skin allograft”

Prepared by authors: I.En Wu, Madonna Rica Anggelia, Sih-Yu Lin, Chiao-Yun Chen, I-Ming Chu, Cheng-Hung Lin

General Comments:

This manuscript presented an interesting approach of designing novel extended delivery systems based on thermosensitive hydrogel of methoxy-poly(ethylene glycol)-co-poly(lactid acid)-poly(e-caprolactone) (mPEG-PLCL), as a drug depot. As a model drug was used tacrolimus (FK506) as a common immunosuppressive drug which is able to suppress acute rejection reactions and successfully used to treat the patients after allotransplation. Due to conventional formulation of tacrolimus (oral or injection route) may cause overdose or low patient compliance problems, the authors developed and presented the new formulation. The synthesized mPEG-PLCL was characterized by 1H-NMR, GPC, FTIR and biocompatibility was evaluated by MTT test. The results of in vitro degradation showed that the hydrogels was completely degraded during the 30 days. Obtained results suggest that novel hydrogels possess various advantageous features (in situ gelling, drug release without no burst effect, slow and steady release over extended period of time, improved allograft outcome and majority degraded in 30 days) which  make them favorable for drug delivery application for immunosuppression treatment after allotransplantation. The paper is well written and organized. This topic should be of interest for the community of biomaterials, biomedicine science, and pharmacy. Therefore, I suggest an acceptance for publication of this manuscript.

Author Response

Dear Editor,

We have revised the manuscript according to reviewers’ suggestions and replied their requests when we think the additional work suggested was not necessary. Our English was also proofread by a native English speaker. Our point-to-point response is listed below. The manuscript was resubmitting with revision texts marked in red.

Author Response:

We appreciate the Reviewer’s comments and recommendations. We have checked the minor spell. The revision in the text is shown in red.

Reviewer 2 Report

interesting workdone by the I-En Wu et al. it could be published in gels journal after addressing below comments; 

1. Figrue 1, it should be proposed reaction scheme? 
2. in methodology portion i.e., 2.4, 2.5 procedure and aim of respective technique should be given!
3. porosity, stability analysis of prepared hydrogels should be performed
4. in order to determin the thermal stability, TGA/DSC analysis must be done for better characterization

Author Response

Dear Editor,

We have revised the manuscript according to reviewers’ suggestions and replied their requests when we think the additional work suggested was not necessary. Our English was also proofread by a native English speaker. Our point-to-point response is listed below. The manuscript was resubmitting with revision texts marked in red.

We believe the manuscript now should be able to satisfy reviewers’ requests and meet the standard of the Journal. We are looking forward to hearing from you for your further instruction.

  1. Figrue 1, it should be proposed reaction scheme? 

Author Response:

Thank you for reviewer correction, the figure caption has been modified.

“Figure 1. Synthetic scheme of mPEG-PCL copolymer hydrogel by ring-opening polymerization of e-caprolactone, initiated by mPEG with Sn(Oct) 2 as a catalyst.”

  1. in methodology portion i.e., 2.4, 2.5 procedure and aim of respective technique should be given!

Author Response:

Thank you for reviewer suggestion, the description has been revised at page 3.

2.4. Fourier-Transformed Infrared Spectroscopy (FT-IR)

The functional group identification of mPEG-PLCL hydrogel was done by Fourier-transformed infrared spectroscopy (FT-IR). Hydrogel infrared spectra was performed using an FTIR spectrometer (NicoletTM iS50, Thermo Fisher Scientific, Waltham, MA, USA) equipped with an attenuated total reflectance (ATR) module. The lyophilized hydrogel was placed on the test plate. Spectra were collected at a frequency ranging from 650 to 4000 cm-1 for 32 times, and the resolution was 1 cm-1. The existence of a C=O bond at 1750 cm-1 and C-O bond at 1100 cm-1 was used to collect information on secondary structures and conducted for chemical structure verification.”

2.5. Sol-gel transition phase

The sol-to-gel transition behavior of mPEG-PLCL prepared at 5 to 25 wt% concentrations was investigated using a vial inverted method in 2 ml test tube at a temperature range of 10–50 ◦C with a temperature increment of 1 ◦C via dry bath incubator. The sample was allowed to equilibrate for 10 min at each temperature. The gel state was determined when the solution stopped flowing while inverted and gently agitated.”

  1. porosity, stability analysis of prepared hydrogels should be performed

Author Response:

From the Fig.4 B (4) photograph in the manuscript, the opaque gel state shown absorbed all the water, i.e. no water was observed clinging to the gel surface. And the composition of that gel is 25 wt% of polymers in water. So, it can be deduced the water content is 75 wt% and the porosity is approximately at that level, 75%, assuming the density differences between water and the hydrated polymers were not great. And as reported in the degradation results, the hydrogel degraded into lower molecular weight substances in about 30 days in water solution, in vitro, while completely disappeared in vivo in 30 days. So the stability of the gels was reported in the paper. For the shelf life of the lyophilized powder of the polymers, it is observed that it remained functional for over a year in storage.

  1. in order to determine the thermal stability, TGA/DSC analysis must be done for better characterization

Author Response:

The question of thermostability of the hydrogels may be an important issue for material characterization. However, in the case of the current thermosensitive hydrogels, an upper limit of temperature for the existence of gel state was mentioned in the paper. Above temperature 46C (for example, see Fig.4 (B5)), phase separation occurs, and the polymers precipitate out. We understand DSC measurements may also be used to confirm polymer composition and their integrity at higher temperatures. The NMR and FTIR data can confirm the composition of our polymers, and for our limited medical applications at body temperatures, DSC measurements to higher temperature ranges seem not to be absolutely necessary.
